# Floristic Similarities between the Lichen Flora of Both Sides of the Drake Passage: A Biogeographical Approach

**DOI:** 10.3390/jof10010009

**Published:** 2023-12-22

**Authors:** Leopoldo G. Sancho, Ana Aramburu, Javier Etayo, Núria Beltrán-Sanz

**Affiliations:** 1Faculty of Pharmacy, Section of Botany, Complutense University, Pl. de Ramón y Cajal, s/n, 28040 Madrid, Spain; anarambu@ucm.es (A.A.); nbeltran@ucm.es (N.B.-S.); 2Calle Navarro Villoslada 16, 3º dcha., Navarra, 31003 Pamplona, Spain; jetayosa@educacion.navarra.es

**Keywords:** Antarctica, biogeography, lichens, Navarino Island

## Abstract

This paper analyses the lichen flora of Navarino Island (Tierra del Fuego, Cape Horn Region, Chile), identifying species shared with the South Shetland Islands (Antarctic Peninsula). In this common flora, species are grouped by their biogeographic origin (Antarctic–subantarctic endemic, austral, bipolar, and cosmopolitan), their habitat on Navarino Island (coastal, forest, and alpine), their morphotype (crustaceous, foliaceous, fruticulose, and cladonioid), and the substrate from which they were collected (epiphytic, terricolous and humicolous, and saxicolous). A total of 124 species have been recognised as common on both sides of the Drake Passage, predominantly bipolar, crustaceous, and saxicolous species, and with an alpine distribution on Navarino Island. The most interesting fact is that more than 30% of the flora is shared between the southern tip of South America and the western Antarctic Peninsula, which is an indication of the existence of a meridian flow of propagules capable of crossing the Antarctic polar front.

## 1. Introduction

The Antarctic flora is usually viewed as being particularly isolated and unique, with special attention given to its high proportion of endemic species in both mosses and lichens [1,2,3]. However, when considering the overall diversity of Antarctica, it is clear that this region is neither isolated nor depauperate [4]. Although the Antarctic continent is far away from other land masses, the Antarctic Peninsula and adjacent islands are only 900 km away from the southern tip of South America, where high mountains, permanent snow cover, and glaciers provide an environment more similar to that of Antarctica than most of the subantarctic islands scattered across the south of the trans-hemispheric oceans. Both sides of the Beagle Channel are dominated by steep slopes, often over a thousand metres high, well above the tree line. An important part of this area, including Navarino Island, is protected and managed under the umbrella of the Cape Horn Biosphere Reserve [5].

Nevertheless, the powerful Antarctic Polar Front (APF) running just between Cape Horn and the Antarctic Peninsula is a major barrier to the latitudinal gradient of dispersal of plant propagules [2]. To what extent the APF has isolated the lichen flora of these two regions is still an open question with important implications in a global change scenario.

The Cape Horn region and the western Antarctic Peninsula share a climate dominated by westerly winds that bring moisture from the Pacific Ocean and reduce seasonal temperature variations. The main consequences are high annual precipitation rates, relatively mild winters, and cool summers. Very often, this precipitation occurs in the form of snow which only partially melts during the summer at altitudes of several hundred metres above sea level. A sharp tree line can only be observed at an altitude of about 450–500 m; above this altitude, pulvinar hemispherical shrubs are interspersed with alpine meadows and cryptograms, that become dominant from about 600 m high [5]. At this altitude, the mean annual air temperature is comparable to that measured at the Antarctic stations on the South Shetland Islands [6].

It is not surprising, therefore, to find a cryptogamic tundra in the Beagle Channel mountains that is physiognomically very analogous to that of the western Antarctic Peninsula. Something similar could be said of the coast, where both shingle beaches and cliffs support lichen communities that mirror those of the South Shetland Islands. On the other hand, both sides of the Drake Passage show an equivalent floristic richness of lichen flora which amounts to about 300–400 species in each of these regions. However, it remains an open question whether this is just a physiognomical affinity and how many species are actually shared between the two sites. The aim of this paper is to analyse the recently published lichen flora of Navarino Island [7], one of the largest and highest islands on the southern side of the Beagle Channel, to determine the degree of affinity with the West Antarctic region (more specifically, with the South Shetland Islands). Within the set of common species, we aim to quantify their distribution along the main altitudinal ranges present on Navarino Island, taking into account their morphology, substrate, habitat, and biogeographical origin.

## 2. Materials and Methods

### 2.1. Common Species between Navarino Island and the West Antarctic Region

We carried out a floristic analysis that benefits from several studies initiated by our group over 30 years ago in the West Antarctic Peninsula region [8,9,10] and on Navarino Island [7,11]. To identify lichen species shared between said regions, the Antarctic lichen flora [12] and the lichen flora of King George Island [13] were also examined. Nomenclature conforms almost entirely to that used on the website of the Consortium of Lichen Herbaria (CNALH) and the Global Biodiversity Information Facility (GBIF).

The set of common species between Navarino Island and maritime Antarctica has been divided into four biogeographical groups according to Sancho et al. 1999 [9], Oevstedal and Lewis Smith 2001 [12], Olech 2004 [13], and Soechting et al. 2004 [10]: Antarctic–subantarctic endemic, which includes the subantarctic islands and the Cape Horn region; austral, i.e., Southern Hemisphere distribution; bipolar, species distributed throughout both polar regions as well as in the high mountains of both hemispheres; and cosmopolitan, species with a broad distribution around the globe. Shared specimens have also been categorised according to a simplified version of the three main altitudinal belts of Navarino Island: coastal area, forest, and alpine region. The coastal habitat consists of trees close to the sea, pebble beaches, cliffs, and rocky intertidal zones; the forest (from around a hundred-meter distance from the shore up to about 500 m a.s.l.) includes both evergreen subantarctic Magellanic rainforests and deciduous forests; and the alpine or high Andean habitat (above 500 m a.s.l.) comprises low shrubs, cushion plants, and rocks scattered beyond the timberline. Each species of the Antarctic–Fuegian flora has been assigned to the substrate(s) on which it was found on Navarino Island. Lichens growing on trunks, branches, wood, and bark on trees were considered epiphytic, as were those found on the branches of cushion bog shrubs such as *Bolax gummifera* and *Empetrum rubrum* in the alpine region. Specimens growing on rocks were classified as saxicolous, while those that developed on soil, humus, grasses, mosses, liverworts, and/or other lichens were seen as terricolous. It should be noted that substrate type and habitat are not mutually exclusive as some species were found living on different substrates across more than one habitat. All possibilities were taken into consideration for our study.

### 2.2. Similarity Indexes

Similarity indexes (SIs) were calculated between the lichen flora from Navarino Island and that of the South Shetland Islands; for both, there was a substantial number of published lists from detailed surveys already available [7,9,10,11,12,13]. The SIs were calculated for the site pair as follows:$$\frac{2\Sigma nc}{\Sigma n1+\Sigma n2}$$
where *nc* is the number of species common between the sites and Σ*n*1 and Σ*n*2 are the total number of species at each site.

### 2.3. Distributional Pattern Analysis

To evaluate the relevance of distributional patterns on the species shared by maritime Antarctica and Navarino Island, we designed a contingency analysis in which species occurrence was calculated in accordance with biogeographical origin, habitat, and substrate type. Since the table generated was a three-way contingency table, a Cochran–Mantel–Haenszel test (CMH test) was used to determine whether there were any associations between variables. Before performing the test, the assumption of homogeneous odd ratios across the levels of each variable was checked using the Woolf test from the R package ‘vcd’ [14]. A post hoc analysis for the CMH test was carried out with the R package ‘rcompanion’ [15]. Additionally, an analysis of the two-way contingency tables was performed using Pearson’s chi-square test of independence for the tables built according to distribution and habitat on one hand and to habitat and substrate type on the other. For the two-way contingency table between distribution and substrate type, a G-test of independence was implemented using the R package ‘DescTools’ (version 0.99.30) [16]. The post hoc analysis was conducted as a pairwise test using the R package ‘rcompanion’ (version 4.1.0) [15]. *p* < 0.01 was considered statistically significant for all tests performed. Plots were created using the R package ‘ggplot2′ (version 3.4.4) [17].

## 3. Results

### 3.1. Common Species between Navarino Island and the West Antarctic Region and Similarity Indexes

Overall, 124 out of the 411 lichen species listed from Navarino Island are shared with the South Shetland Islands, which means that 30.2% can be considered Antarctic–Fuegian flora. Within this common flora, the dominant biogeographical group is the bipolar group (63 species), followed by the cosmopolitan group (23 species), the endemic group (21 species), and lastly, austral group (17 species). The alpine region of Navarino Island harbours 43.8% of the flora shared with Antarctica, while the forest represents only 23.8%. The remaining 32.5% of this common flora corresponds to the coastline (Table 1). Otherwise, the crustaceous morphotype is the most abundant, followed by foliaceous, cladonioid, and fruticose morphotypes (Table 2).

On the other hand, since the lichen flora of the South Shetland Islands accounts for 313 species (combining the floras of Livingston Island and King George Island), 39.6% of them are present on the southern tip of South America. If we extract from the 411 Navarino lichen species those that appear to be exclusively epiphytic—around 115—and those whose presence in Antarctica is highly improbable, we would be left with a pool rather similar to the floristic richness of maritime Antarctica. The degree of similarity would then be remarkable on both sides of the Drake Passage. The SI considering the whole Navarino lichen flora with respect to the South Shetland Islands is 0.343, and it is 0.407 if we discard the exclusively epiphytic species.

In the Antarctic–Fuegian flora, *Cladonia* and *Rhizocarpon* are the genera with the highest numbers of species. Interestingly, no species belonging to the important genus *Buellia* are shared between both sides of the Drake Passage. Nor, of course, are the more typical forest species of the genera *Nephroma*, *Sticta*, and *Pseudocyphellaria*. A few species, such as *Poeltidea perusta* and *Rhizocarpon geographicum,* are found across all habitats on Navarino Island. Most of the shared species show sexual reproduction, but asexual propagules (soredia, isidia, and thalloconidia) are also common.

### 3.2. Distributional Pattern Analysis

The three-way contingency table (Table 3) showed that the alpine habitat had a high proportion of terricolous species with bipolar distribution (28.6%; see Table 3 and Figure 1). The alpine saxicolous and epiphytic species with bipolar distribution were also present in higher proportions with respect to the other three biogeographical origins considered in our study (23.8 and 9.5%, respectively). In the coastal habitat, bipolar saxicolous species were predominant (25.0%). Saxicolous specimens with endemic and cosmopolitan distributions are also notoriously present (16.7 and 13.3%, respectively). In the forest habitat, bipolar terricolous species are the most abundant ones (23.3%), followed by cosmopolitan terricolous (18.6%) and bipolar saxicolous (14.0%) species.

Regarding substrate, saxicolous lichens were dominant for all biogeographical groups except for the cosmopolitan group, for which the terricolous lichens prevailed. Epiphytes represent the smallest group in all cases (Figure 2).

The Woolf test for the two-way contingency table showed homogeneous odds ratios across the levels of each variable with a *p*-value of 0.6, thus satisfying the assumptions needed to perform the CHM test. The latter indicated that habitat and substrate are dependent on one another at a given distribution level (*p* = 0.00009). This dependence occurred in particular at the cosmopolitan distribution level, as shown by the post hoc analysis (*p* = 0.004), which can be explained by the fact that there are barely any epiphytic lichens with a cosmopolitan distribution in the Antarctic–Fuegian flora. Furthermore, there are no saxicolous representatives within this biogeographical group in the forest habitat. Most of the cosmopolitan specimens are either saxicolous in coastal areas or terricolous in forests.

The G-test of independence performed for the two-way contingency table built according to distribution and substrate type presented no statistically significant differences across levels (*p* = 0.131). The chi-square tests carried out for the tables that took into account habitat and distribution and habitat and substrate type showed that only the latter had any statistically significant differences (*p* = 0.0001). This finding corroborated the result given by the CMH-test. The post hoc analysis indicated that the dependence found between these two variables was mainly due to the distinct proportion of saxicolous versus terricolous species throughout the habitats (*p* = 0.0001), particularly in the coast and forest; while saxicolous lichens are mainly found in coastal areas, terricolous ones prefer forests. Epiphytic specimens, however, are distributed in a more uniform way across the three ambient types (Table 4).

## 4. Discussion

Antarctica is the most isolated continent on Earth. The powerful Antarctic Polar Front (APF) is a significant barrier to both marine and terrestrial organisms. However, there is evidence that this barrier can be overcome [4]. This is clearly the case for the lichen flora on both sides of the Drake Passage. The proportion of lichen species from the South Shetland Islands that are also found on Navarino Island is roughly 40%, and the same is true for the flora of Navarino Island if we extract the approximately 115 exclusively epiphytic species from its more than 400 known species [7] (ca. 42%). The SI between Navarino Island and the South Shetland Islands is between 0.342 and 0.407 (depending on whether the epiphytic species are included or not, respectively). This is similar to the SI between the large islands of the Canadian Arctic Archipelago [18], amongst which there are no significant geographical barriers. Additionally, active genetic exchange in both north–south and south–north directions has been reported for some species from the genera *Usnea* and *Mastodia* [19,20,21].

Among the common species surpassing the APF, the bipolar biogeographical group stands out, accounting for 50.8% of the Antarctic–Fuegian flora (Table 1). Most of the species belonging to this group have their largest distribution in the vast Arctic and high mountain areas of the Northern Hemisphere, but in many cases, their centre of diversity is yet to be determined. The cosmopolitan group (18.6%) includes those species that are able to cross all geographical and climatic barriers to then grow on any part of our planet, including Antarctica. Both the bipolar and the cosmopolitan percentages are very similar to those reported by Ochyra et al. (2008) [22] for Antarctic mosses. The austral biogeographical group (13.7%) includes some species that may have originated on the old continent of Gondwana, as did many vascular plants [23,24] and bryophytes [25]. Finally, the Antarctic–subantarctic endemic group (16.9%) should be considered to be composed of pre-Pleistocene survivors of the ice ages that somehow found refuge in ice-free areas, mainly on rocky substrates in coastal regions or nunataks [23,26,27]. Interestingly, some cosmopolitan species, such as *Collema tenax* (Sw.) Ach. and *Lecidea lapicida* (Ach.) Ach., and bipolar species, such as *Caloplaca tirolensis* Zahlbr. and *Dermatocarpon polyphyllizum* (Nyl.) Blomb. & Forssell, that live on the South Shetland Islands are not found on Navarino Island, evidencing more complex colonisation processes than just a latitudinal gradient of dispersal.

The evidence of APF crossings is also a warning of the possible arrival of invasive species in a global warming scenario. Newly arrived terrestrial species have already been documented in maritime Antarctica, but no lichens have been documented so far [28]. However, natural dispersal events are no longer necessary to bring invasive species to Antarctica. It has been estimated that 40% of scientific visitors and 20% of tourists carry plant propagules from northern regions [29].

In terms of habitat distribution, it is not surprising that the highest number of species from Navarino Island shared with maritime Antarctica is found in the alpine habitat and that the lowest number is found in the forest habitat. It is more interesting to note that the coast has a relatively high percentage of affinities with Antarctica. This may be related to the very recent onset of forests in this region, dated to the middle Holocene, between 6000 and 5000 years before the present in Tierra del Fuego [30] and slightly earlier on Isla de los Estados [31]. The encroachment of the *Nothofagus* forest separated what must have been a continuous tundra from the sea to the peaks, similar to what is found in the ice-free areas of Antarctica today. In any case, this is a region of strong climatic oscillations driven by changes in latitude and the strength of westerly winds. Incidentally, the maximum glacial extent of the Holocene occurred only about 800 years ago, well before the Little Ice Age, probably as a result of a southward shift of these dominant winds [32]. Vegetation must adapt to these continuous environmental changes, using the altitudinal gradient as a buffer against the effects of climatic oscillations.

It is also understandable that the preferred substrates for this Antarctic–Fuegian flora are rocks, and that epiphytes are the least common species type. There is a considerable number of species growing on soil or mosses, all of which would be considered components of a cryptogamic tundra in the strict sense. Most of these species belong to the genus *Cladonia* which, by far, accounts for the largest number of species in this Antarctic–Fuegian flora. Our knowledge of this genus is particularly good thanks to a study carried out by Burgaz and Raggio in 2007 [11]. Further monographic work on key genera is likely to add new species to our catalogue of Navarino Island.

Regarding morphology, it is striking that very few species with a cyanobacterial photosymbiont are shared between the two sites. For example, the genus *Placopsis*, which is ubiquitous in the Southern Hemisphere, shows no common species on either side of the Drake Passage, while there are species such as *P. perrugosa* that are shared between South America and New Zealand [33] and two others that are restricted to Antarctica (*P. antarctica* and *P. contortuplicata*). The same can be said for some large green algae lichen genera, such as *Buellia*, that are present and abundant on both sides of the Drake Passage and yet have no shared species. This suggests that the barrier between Antarctica and the rest of the world could be ecological rather than geographical. The cold Antarctic summer, with average temperatures only slightly above freezing, may be the main limiting factor for the establishment and germination of propagules from the north. Some maritime Antarctic endemisms, such as *Himantormia lugubris*, show an extreme dependence on these cold and wet summer conditions [34]. In a warming scenario, these endemic species would be expected to decline in favour of other more widely distributed taxa [35]. Long-term studies at numerous sites on the Antarctic Peninsula and adjacent islands are essential to detect changes in floral composition that can only be hypothesised at this stage.

## Figures and Tables

**Figure 1 jof-10-00009-f001:**
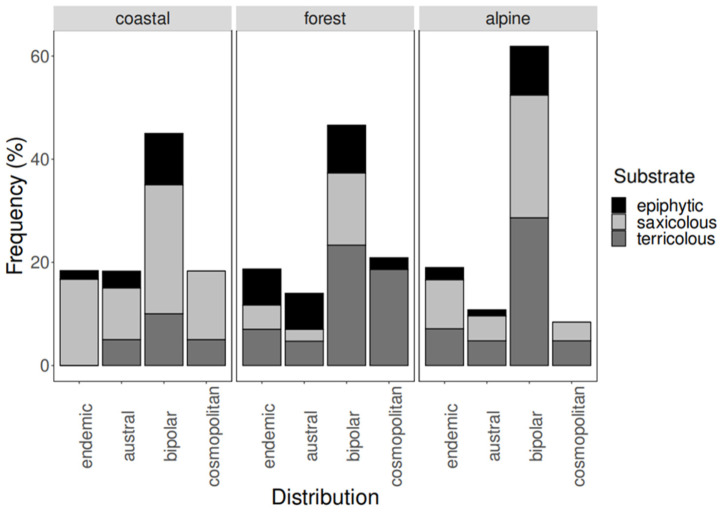
Frequency plot of the shared species between Navarino Island and the South Shetland Islands according to their distribution, habitat, and substrate type on Navarino Island. Relative species abundance is expressed with respect to habitat.

**Figure 2 jof-10-00009-f002:**
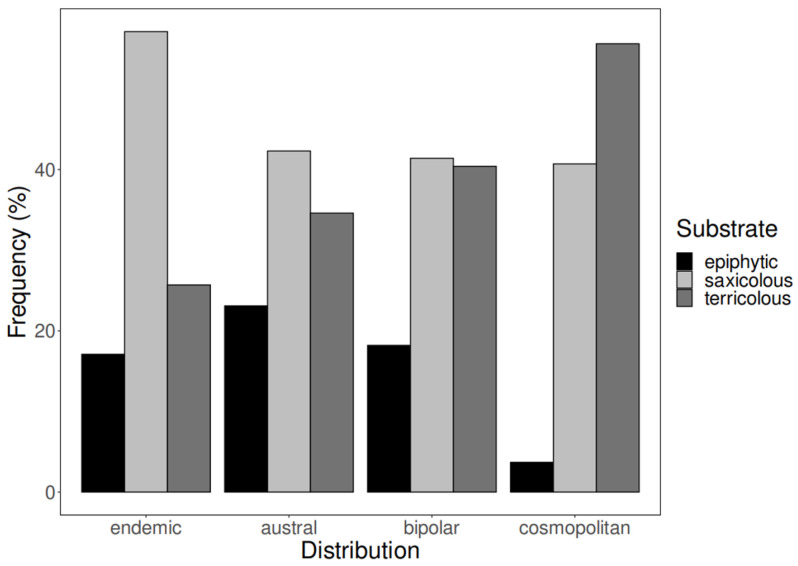
Frequency plot of the shared species between Navarino Island and the South Shetland Islands according to their distribution and substrate type on Navarino Island.

**Table 1 jof-10-00009-t001:** Shared species between Navarino Island and the South Shetland Islands. Endemic stands for Antarctic–subantarctic endemic.

Distribution	Species	Habitat			Substrate Type		
		Coastal	Forest	Alpine	Epiphytic	Saxicolous	Terricolous
bipolar	*Acarospora badiofusca* (Nyl.) Th. Fr.	+				+	
cosmopolitan	*Agonimia tristicula* (Nyl.) Zahlbr.	+				+	
bipolar	*Amandinea punctata* (Hoffm.) Coppins & Scheid.	+	+		+		
bipolar	*Arthrorhaphis citrinella* (Ach.) Poelt			+			+
endemic	*Austroplaca ambitiosa* (Darb.) Søchting, Frödén & Arup	+				+	
endemic	*Austroplaca cirrochrooides* (Nyl.) Søchting, Frödén & Arup s.lat.	+				+	
endemic	*Austroplaca millegrana* (Müll. Arg.) Søchting, Frödén & Arup s.lat.	+				+	
cosmopolitan	*Baeomyces rufus* (Huds.) Rebent.			+			+
austral	*Bibbya bullata* (Meyen & Flot.) Kistenich, Timdal, Bendiksby & S. Ekman	+			+	+	+
bipolar	*Caloplaca phaeocarpella* (Nyl.) Zahlbr			+	+		
cosmopolitan	*Candelariella vitellina* (Ehrh.) Müll. Arg.	+				+	
bipolar	*Carbonea vorticosa* (Flörke) Hertel		+	+		+	
bipolar	*Catapyrenium cinereum* (Pers.) Körb.			+			+
bipolar	*Cetraria aculeata* (Schreb.) Fr.			+			+
bipolar	*Cetraria islandica* (L.) Ach.			+			+
bipolar	*Chrysothrix chlorina* (Ach.) J. R. Laundon	+				+	
austral	*Cladia aggregata* (Sw.) Nyl.	+		+			+
bipolar	*Cladonia asahinae* J.W. Thompson			+	+		+
bipolar	*Cladonia bellidiflora* (Ach.) Schaerer		+	+			+
bipolar	*Cladonia borealis* Stenroos		+	+			+
bipolar	*Cladonia carneola* (Fr.) Fr.			+			+
endemic	*Cladonia cervicornis* spp. mawsonii (C.W. Dodge) S. Stenroos & Ahti			+			+
cosmopolitan	*Cladonia chlorophaea* (Flörke ex Sommerf.) Spreng.	+	+	+			+
cosmopolitan	*Cladonia cornuta* (L.) Hoffm.	+	+				+
cosmopolitan	*Cladonia fimbriata* (L.) Fr.		+				+
bipolar	*Cladonia gracilis* (L.) Willd.			+		+	+
endemic	*Cladonia lepidophora* Ahti & Kashiw.		+				+
bipolar	*Cladonia mitis* Sandst.		+	+			+
endemic	*Cladonia novochlorophaea* (Sipman) Brodo & Ahti		+				+
bipolar	*Cladonia phyllophora* Hoffm.		+				+
bipolar	*Cladonia pleurota* (Flörke) Schaerer		+				+
cosmopolitan	*Cladonia pocillum* (Ach.) Grognot		+				+
cosmopolitan	*Cladonia pyxidata* (L.) Hoffm.		+				+
bipolar	*Cladonia rangiferina* (L.) Weber ex Wigg.		+	+			+
austral	*Cladonia sarmentosa* (Hook f. & Taylor) C.W. Dodge			+			+
bipolar	*Cladonia scabriuscula* (Delise) Nyl.		+			+	
cosmopolitan	*Cladonia squamosa* (Scop.) Hoffm.		+				+
austral	*Cladonia subsubulata* Nyl.		+				+
bipolar	*Cladonia subulata* (L.) F.H. Wigg.		+				+
bipolar	*Cladonia sulphurina* (Michaux) Fr.		+	+			+
austral	*Cladonia ustulata* (Hook. F. & Taylor) Leight.	+	+				+
austral	*Coccotrema cucurbitula* (Mont.) Müll. Arg.		+	+	+	+	
austral	*Coelopogon epiphorellum* (Nyl.) Brusse & Kärnefelt	+	+		+	+	
bipolar	*Cystocoleus ebeneus* (Dillwyn) Twaites	+		+		+	+
bipolar	*Frutidella caesioatra* (Schaer.) Kalb			+			+
austral	*Gondwania sublobulata* (Nyl.) S.Y. Kondr., Kärnefelt, Elix, A. Thell, J. Kim, M.-H. Jeong, N.-N. Yu, A.S. Kondr. & Hur	+				+	
bipolar	*Gowardia nigricans* (Ach.) Halonen			+	+		+
endemic	*Haematomma erythromma* (Nyl.) Zahlbr.	+		+		+	
cosmopolitan	*Hydropunctaria maura* (Wahlenb.) C. Keller, Gueidan & Thüs	+				+	
bipolar	*Hypogymnia lugubris* var. lugubris (Pers.) Krog			+	+	+	
bipolar	*Japewia tornoensis* (Nyl.) Tønsberg			+	+		+
bipolar	*Lecanora epibryon* (Ach.) Ach.			+			+
cosmopolitan	*Lecanora flotoviana* Spreng.	+				+	
bipolar	*Lecanora intricata* (Ach.) Ach.			+		+	
endemic	*Lecanora physciella* (Darb.) Hertel			+	+	+	+
bipolar	*Lecanora polytropa* (Hoffm.) Rabenh.			+		+	
bipolar	*Lecidea atrobrunnea* (Ram. ex Lam. et DC.) Schaerer			+		+	
bipolar	*Lecidella elaeochroma* (Ach.) M. Choisy	+			+		
bipolar	*Lecidella patavina* (A. Massal.) Knoph & Leuckert	+				+	
bipolar	*Lecidella stigmatea* (Ach.) Hertel & Leuckert	+				+	
bipolar	*Lecidella wulfenii* (Hepp) Körb.	+			+		
bipolar	*Lecidoma demissum* (Rustr.) Gotth. Schneid. & Hertel			+			+
bipolar	*Lepraria caesioalba* (B. de Lesd.) J.R. Laundon	+	+	+		+	+
endemic	*Leptogium menziesii* (Ach.) Mont.		+	+			+
endemic	*Leptogium puberulum* Hue			+		+	
bipolar	*Massalongia patagonica* Kitaura & Lorenz	+	+				+
bipolar	*Mastodia tessellata* (Hook. f. & Harv.) Hook. f. & Harv.	+				+	
bipolar	*Megalaria grossa* (Pers. ex Nyl.) Hafellner	+		+	+		
bipolar	*Megaspora verrucosa* (Ach.) Hafellner & V. Wirth	+			+		+
cosmopolitan	*Melanohalea elegantula* (Zahlbr.) O. Blanco, A. Crespo, Divakar, Essl., D. Hawksw. & Lumbsch		+		+		
endemic	*Melanohalea ushuaiensis* (Zahlbr.) O. Blanco et al.	+	+		+	+	
endemic	*Menegazzia magellanica* R. Sant.		+		+	+	
bipolar	*Micarea incrassata* Hedl.	+					+
bipolar	*Mycobilimbia hypnorum* (Lib.) Kalb & Hafellner			+			+
austral	*Notoparmelia cunninghamii* (Cromb.) A. Crespo, Ferencová & Divakar		+		+		
endemic	*Ochrolechia antarctica* (Müll. Arg.) Darb.	+		+		+	
bipolar	*Ochrolechia frigida* (Sw.) Lynge	+		+			+
bipolar	*Pannaria hookeri* (Borrer ex Sm.) Nyl.			+		+	+
cosmopolitan	*Parmelia saxatilis* (L.) Ach.	+		+		+	
cosmopolitan	*Peltigera didactyla* (With.) J.R. Laundon		+				+
cosmopolitan	*Peltigera rufescens* (Weiss) Humb.		+				+
endemic	*Peltularia fuegiana* Henssen & P.M. Jørg.		+	+	+	+	+
endemic	*Pertusaria spegazzinii* Müll. Arg.	+				+	
bipolar	*Phaeophyscia endococcina* (Körb.) Moberg	+				+	
cosmopolitan	*Physcia caesia* (Hoffm.) Fürnr.	+				+	
cosmopolitan	*Physcia dubia* (Hoffm.) Lettau	+				+	
bipolar	*Physconia muscigena* (Ach.) Poelt	+				+	
cosmopolitan	*Placidium squamulosum* (Ach.) O. Breuss	+					+
endemic	*Poeltidea perusta* (Nyl.) Hertel & Hafellner	+	+	+		+	
bipolar	*Polycauliona candelaria* (L.) Frödén, Arup & Søchting	+	+	+	+	+	
bipolar	*Pseudephebe minuscula* (Nyl.) Brodo & D. Hawksw.			+		+	
endemic	*Psoroma antarcticum* Hong & Elvebakk			+			+
endemic	*Psoroma cinnamomeum* Malme			+	+	+	+
austral	*Psoroma fruticulosum* James & Henssen			+		+	
cosmopolitan	*Psoroma hypnorum* (Vahl) Gray			+			+
endemic	*Ramalina terebrata* Hook. & Taylor	+				+	
bipolar	*Rhizocarpon geminatum* Körb.	+		+		+	
bipolar	*Rhizocarpon geographicum* (L.) DC.	+	+	+		+	
bipolar	*Rhizocarpon polycarpum* (Hepp) Th. Fr.			+		+	
bipolar	*Rinodina olivaceobrunnea* C.W. Dodge & G.E. Baker			+	+		+
austral	*Rinodina peloleuca* (Nyl.) Müll. Arg.	+				+	
cosmopolitan	*Sarcogyne privigna* (Ach.) A. Massal.	+				+	
austral	*Siphulastrum mamillatum* (Hook. F. & Taylor) D.J. Galloway			+			+
cosmopolitan	*Sphaerophorus globosus* (Huds.) Vainio			+			+
bipolar	*Sporastastia testudinea* (Ach.) A. Massal.			+		+	
bipolar	*Stereocaulon alpinum* Laurer	+		+			+
austral	*Stereocaulon glabrum* (Müll. Arg.) Vain.			+			+
cosmopolitan	*Tephromela atra* (Huds.) Hafellner			+		+	
bipolar	*Toniniopsis bagliettoana* (A. Massal. & De Not.) Kistenich & Timdal			+	+		
bipolar	*Trapeliopsis granulosa* (Hoffm.) Lumbsch		+		+		
bipolar	*Tremolecia atrata* (Ach.) Hertel			+		+	
bipolar	*Umbilicaria decussata* (Vill.) Zahlbr.			+		+	
bipolar	*Umbilicaria nylanderiana* (Zahlbr.) H. Magn.			+		+	
bipolar	*Umbilicaria umbilicarioides* (Stein.) Krog & Swinscow			+		+	
austral	*Usnea aurantiaco-atra* (Jacq.) Bory			+		+	
bipolar	*Usnea* cf. *sphacelata* R. Br.	+	+	+	+	+	
endemic	*Usnea subantarctica* F.J. Walker			+		+	
austral	*Usnea trachycarpa* (Stirton) Müll. Arg.			+		+	
endemic	*Verrucaria dispartita* Vain.	+				+	
austral	*Verrucaria durietzii* I.M. Lamb	+				+	
bipolar	*Verrucaria halizoa* Leight.	+				+	
bipolar	*Verrucaria margacea* (Wahlenb. in Ach.) Wahlenb.	+				+	
austral	*Verrucaria tesselatula* Nyl.	+				+	
cosmopolitan	*Xanthoria elegans* (Link) Th. Fr.			+		+	

The symbol + indicates which habitat and substrate type categories apply to each species.

**Table 2 jof-10-00009-t002:** Abundance of the main morphotypes of the shared species between Navarino Island and the South Shetland Islands.

Morphology	Absolute Abundance	Relative Abundance
crustose	56	45.2%
foliose	28	22.6%
fruticose	16	12.9%
cladonoid	24	19.3%

**Table 3 jof-10-00009-t003:** Three-way contingency table of the shared species between Navarino Island and the South Shetland Islands according to their distribution, habitat, and substrate type on Navarino Island. Relative species abundance is expressed with respect to habitat.

Habitat	Substrate Type	Distribution			
		Endemic	Austral	Bipolar	Cosmopolitan
coastal	epiphytic	1.7%	3.3%	10.0%	0.0%
	saxicolous	16.7%	10.0%	25.0%	13.3%
	terricolous	0.0%	5.0%	10.0%	5.0%
forest	epiphytic	7.0%	7.0%	9.3%	2.3%
	saxicolous	4.7%	2.3%	14.0%	0.0%
	terricolous	7.0%	4.7%	23.3%	18.6%
alpine	epiphytic	2.4%	1.2%	9.5%	0.0%
	saxicolous	9.5%	4.8%	23.8%	3.6%
	terricolous	7.1%	4.8%	28.6%	4.8%

**Table 4 jof-10-00009-t004:** Two-way contingency table of the shared species between Navarino Island and the South Shetland Islands according to their habitat and substrate type on Navarino Island.

Substrate Type	Habitat		
	Coastal	Forest	Alpine
epiphytic	15.0%	25.6%	13.1%
saxicolous	65.0%	20.9%	41.7%
terricolous	20.0%	53.5%	45.2%

## Data Availability

Lichen samples used are deposited in the Pharmacy Herbarium MAF of the Complutense University of Madrid.

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
