# Peer review of "Floristic Similarities between the Lichen Flora of Both Sides of the Drake Passage: A Biogeographical Approach"

_jof, 2023, doi:10.3390/jof10010009_

Round 1

Reviewer 1 Report

Comments and Suggestions for Authors

I found this an interesting paper. It is well written in general and I only have minor comments.

Could the authors say a little more in the Introduction about why they think this study is useful? Perhaps a little more about the effectiveness of the Drake Passage as a dispersion/migration barrier. For example, the Antarctic Polar Front is mentioned in the Discussion but could also be introduced here.

I wonder whether it would be of interest to perhaps say a little more about Antarctic species that have not made there way onto Navarino Island. This has been touched on briefly in the final paragraph of the Discussion. For example, are there some cosmopolitan species that occur on the peninsula that are absent on Nararino?  Are there any trends amongst propagule types e.g. is there any evidence that species that produce soredia migrate more or less effectively?

Line 26: ‘depauperated’ is correct but I ‘depauperate’ might be better

Line 41: ‘above which’?

Lines 82-84: possible rewrite: ‘…….epiphytic, as were those found on branches of cushion bog shrubs such as Bolax gummifera and Empetrum rubrum in the alpine region.’

Lines 106-107: consider re-writing.

Table 1: italicise ‘Gondwana sublobulata’?

Line 133: ‘appear’

Line 252: ‘such as Buellia, that are present on both sides of the Drake Passage,’ But not on Navarino Island? Do you need to state this for clarity?

Lines 253-254: ‘This suggests that the barrier between Antarctica and the rest of the world is eco- logical rather than geographical.’ Would it be better to say ‘can be ecological’?

Author Response

All changes and suggestions made by the review have been taken into account. These changes are specified below:

Comments and Suggestions for Authors

I found this an interesting paper. It is well written in general and I only have minor comments.

Could the authors say a little more in the Introduction about why they think this study is useful? Perhaps a little more about the effectiveness of the Drake Passage as a dispersion/migration barrier. For example, the Antarctic Polar Front is mentioned in the Discussion but could also be introduced here.

To address this point, a short paragraph has been inserted in the Introduction

I wonder whether it would be of interest to perhaps say a little more about Antarctic species that have not made there way onto Navarino Island. This has been touched on briefly in the final paragraph of the Discussion. For example, are there some cosmopolitan species that occur on the peninsula that are absent on Nararino?  Are there any trends amongst propagule types e.g. is there any evidence that species that produce soredia migrate more or less effectively?

Indeed the review raises an important issue. The species have been revised again and a few lines have been added in Discussion accordingly.

Line 26: ‘depauperated’ is correct but I ‘depauperate’ might be better

Accepted and done.

Line 41: ‘above which’?

Accepted and done.

Lines 82-84: possible rewrite: ‘…….epiphytic, as were those found on branches of cushion bog shrubs such as Bolax gummifera and Empetrum rubrum in the alpine region.’

Accepted and done.

Lines 106-107: consider re-writing.

Accepted and done. Accordingly other few changes have been made in Material and Methods to improve the understanding.

Table 1: italicise ‘Gondwana sublobulata’?

Accepted and done.

Line 133: ‘appear’

Accepted and done.

Line 252: ‘such as Buellia, that are present on both sides of the Drake Passage,’ But not on Navarino Island? Do you need to state this for clarity?

Accepted and done.

Lines 253-254: ‘This suggests that the barrier between Antarctica and the rest of the world is eco- logical rather than geographical.’ Would it be better to say ‘can be ecological’?

Accepted and done.

Reviewer 2 Report

Comments and Suggestions for Authors

The aim of this work is to compare the biogeographical biota of both sides of the Drake Passage from a biogeographical point of view. The work is well structured, as it is an enrichment of a very rich basic floristic information, giving a broader view of the biogeography of an extreme area that could be of interest not only to specialists.
On the basis of this approach, the authors point to a considerable sharing of lichen flora between the Antarctic Peninsula and the southernmost part of South America, supporting the hypothesis that lichen propagules may have flowed between the two areas.
Although these considerations are somewhat speculative because they are based on similarity data of floristic compositions and not on genomic data, the statistical methods used and the arguments put forward by the authors are convincing.
So, all in all, I think this is a very nice, interesting and robust piece of work, which can provide a considerable basis for observation in future climate change scenarios, as the authors rightly envisage.
I also liked the point made by the authors in the introduction, where they mention that it remains an
open question whether the similarity of the lichen biota in the two areas is merely a physiognomic affinity or whether many species are actually shared between the two sites. If there is one aspect of the discussion that could be improved with a little rewording, it would be to make more explicit the conclusions that the authors have drawn from their results. Otherwise, the authors have already done an excellent job.

Author Response

The aim of this work is to compare the biogeographical biota of both sides of the Drake Passage from a biogeographical point of view. The work is well structured, as it is an enrichment of a very rich basic floristic information, giving a broader view of the biogeography of an extreme area that could be of interest not only to specialists.

On the basis of this approach, the authors point to a considerable sharing of lichen flora between the Antarctic Peninsula and the southernmost part of South America, supporting the hypothesis that lichen propagules may have flowed between the two areas.

Although these considerations are somewhat speculative because they are based on similarity data of floristic compositions and not on genomic data, the statistical methods used and the arguments put forward by the authors are convincing.

So, all in all, I think this is a very nice, interesting and robust piece of work, which can provide a considerable basis for observation in future climate change scenarios, as the authors rightly envisage.

I also liked the point made by the authors in the introduction, where they mention that it remains an open question whether the similarity of the lichen biota in the two areas is merely a physiognomic affinity or whether many species are actually shared between the two sites. If there is one aspect of the discussion that could be improved with a little rewording, it would be to make more explicit the conclusions that the authors have drawn from their results. Otherwise, the authors have already done an excellent job.

We are very grateful for the positive comments on this review. We think that the small changes introduced in the Discussion make the conclusions clearer.

Reviewer 3 Report

Comments and Suggestions for Authors

The paper presents an analysis of the affinity of the lichen biota of Navarino Island (South America) with the South Shetland Islands (Antarctic Peninsula). The species' biogeographic origin, habitat, substrate and thallus morphology were taken into account. As a result, the authors showed a high degree of similarity between both biotas, and indicated the dominance of crustaceous, saxicolous and bipolar species. The high share of common species on both analysed regions may indicate the possibility of lichens actively crossing the barrier of the Antarctic Polar Front.

The paper is written correctly and very carefully, I noticed only single errors in the text format in Table 1. The introduction is complete, the methodology used is properly applied, the results are clearly presented and the discussion is exhaustive. To sum up, the work is undoubtedly interesting and suitable for publication.

Author Response

Review 3

The paper presents an analysis of the affinity of the lichen biota of Navarino Island (South America) with the South Shetland Islands (Antarctic Peninsula). The species' biogeographic origin, habitat, substrate and thallus morphology were taken into account. As a result, the authors showed a high degree of similarity between both biotas, and indicated the dominance of crustaceous, saxicolous and bipolar species. The high share of common species on both analysed regions may indicate the possibility of lichens actively crossing the barrier of the Antarctic Polar Front.

The paper is written correctly and very carefully, I noticed only single errors in the text format in Table 1. The introduction is complete, the methodology used is properly applied, the results are clearly presented and the discussion is exhaustive. To sum up, the work is undoubtedly interesting and suitable for publication.

We are very grateful for the positive comments on this review.